# Feature architecture aware phylogenetic profiling indicates a functional diversification of type IVa pili in the nosocomial pathogen *Acinetobacter baumannii*

**Ruben Iruegas**[1], **Katharina Pfefferle**[2], **Stephan Göttig**[3], **Beate Averhoff**[2], **Ingo Ebersberger**[1,4,5]*

1 Applied Bioinformatics Group, Inst of Cell Biology and Neuroscience, Goethe University Frankfurt, Frankfurt am Main, Germany, 2 Molecular Microbiology & Bioenergetics, Institute of Molecular Biosciences, Goethe University Frankfurt, Frankfurt am Main, Germany, 3 Institute for Medical Microbiology and Infection Control, University Hospital, Goethe University, Frankfurt, Germany, 4 Senckenberg Biodiversity and Climate Research Centre (S-BIK-F), Frankfurt am Main, Germany, 5 LOEWE Centre for Translational Biodiversity Genomics (TBG), Frankfurt am Main, Germany

* ebersberger@bio.uni-frankfurt.de

**Data Availability Statement:** The authors confirm that all data underlying the findings are fully available without restriction. All numerical data that

## Abstract

The Gram-negative bacterial pathogen *Acinetobacter baumannii* is a major cause of hospital-acquired opportunistic infections. The increasing spread of pan-drug resistant strains makes *A. baumannii* top-ranking among the ESKAPE pathogens for which novel routes of treatment are urgently needed. Comparative genomics approaches have successfully identified genetic changes coinciding with the emergence of pathogenicity in *Acinetobacter*. Genes that are prevalent both in pathogenic and a-pathogenic Acinetobacter species were not considered ignoring that virulence factors may emerge by the modification of evolutionarily old and widespread proteins.

Here, we increased the resolution of comparative genomics analyses to also include lineage-specific changes in protein feature architectures. Using type IVa pili (T4aP) as an example, we show that three pilus components, among them the pilus tip adhesin ComC, vary in their Pfam domain annotation within the genus *Acinetobacter*. In most pathogenic *Acinetobacter* isolates, ComC displays a von Willebrand Factor type A domain harboring a finger-like protrusion, and we provide experimental evidence that this finger conveys virulence-related functions in *A. baumannii*. All three genes are part of an evolutionary cassette, which has been replaced at least twice during *A. baumannii* diversification. The resulting strain-specific differences in T4aP layout suggests differences in the way how individual strains interact with their host. Our study underpins the hypothesis that *A. baumannii* uses T4aP for host infection as it was shown previously for other pathogens. It also indicates that many more functional complexes may exist whose precise functions have been adjusted by modifying individual components on the domain level.

underlies figures and/or summary statistics is within the manuscript and its Supporting information files. Genomic data analyzed in this study are publicly available in the NCBI Reference Sequence Database (RefSeq). For each selected genome, the corresponding assembly accession is listed in S3 Table. We obtained protein sequences and coding sequences (*.faa and *cds_from_genomic.fna), and genome sequence (*_genomic.fna) files from the database [https://ftp.ncbi.nlm.nih.gov/refseq/release/release-catalog/archive/]. The domain architecture-aware phylogenetic profiles for the Ab ATCC 19606 T T4aP components are available via figshare: https://figshare.com/articles/dataset/_/21964535. A high-resolution version of the tree shown as S7 Fig is available via figshare: https://figshare.com/articles/figure/_/21967694.

**Funding:** This study was supported by a grant by the German Research Foundation (DFG; https://www.dfg.de/) in the scope of the Research Group FOR2251 "Adaptation and persistence of A. baumannii." Grant ID EB-947 285-2/2 to IE, AV 9/7-2 to BA, GO 2491/1-2 to SG. BA additionally acknowledges financial support by the Deutsche Forschungsgemeinschaft (https://www.dfg.de/) via AV 916-2, and IE was further supported by the Research Funding Program Landes-Offensive zur Entwicklung Wissenschaftlich-ökonomischer Exzellenz (LOEWE; https://wissenschaft.hessen.de/forschen/landesprogramm-loewe) of the State of Hessen, Research Center for Translational Biodiversity Genomics (TBG). The funders had no role in study design, data collection and analysis, decision to publish, or preparation of the manuscript.

**Competing interests:** The authors have declared that no competing interests exist.

## Author summary

Type IVa pili (T4aP) are hair-like, extendable, and retractable appendages that many bacteria use for interacting with their environment. Several human pathogens have independently recruited these pili for processes related to host infection, but the modifications necessary to turn T4aP into virulence factors are largely unknown. Here, we studied if and how T4aP components have changed in the nosocomial pathogen *A. baumannii* compared to its largely a-pathogenic relatives in the *Acinetobacter* genus. Most *A. baumannii* isolates have T4aP with a pilus tip adhesin containing a protein domain variant not seen outside the pathogenic clade. This variant appears essential for bacterial motility and contributes to host cell adhesion and natural competence. However, some isolates have T4aP resembling those of largely a-pathogenic species in this genus. This indicates that the way these pili are used during infection processes differs between *A. baumannii* isolates probably as a consequence of niche adaptation. In a broader perspective, our findings highlight that many relevant genetic differences between pathogens and their a-pathogenic relatives emerge only on the domain- and sub-domain level. Thus, existing comparative genomics studies have likely uncovered only the tip of the iceberg of genetic determinants that contribute to *A. baumannii* virulence.

## Introduction

*Acinetobacter baumannii* is a Gram negative nosocomial pathogen. A recent world-wide survey estimated that antimicrobial resistant *A. baumannii* strains were responsible for 10.4% of bacterial infections with fatal outcome in 2019 [1]. The spread of multi- or even pan-resistant *A. baumannii* isolates [2–4] is accompanied by a surge in virulence [4–8], and thus novel therapeutic treatments are necessary for sustainable infection management. Over the past years, experimental studies integrated with comparative genomics analyses have sought to identify genetic determinants of *A. baumannii* virulence [9–21]. The resulting candidates are involved in a broad spectrum of biological processes including NOS and ROS resistance and metabolic adaptation, but some also indicate changes in the way the pathogen interacts with its environment [15,21].

Pili, or 'hair-like' surface appendages, are main mediators of bacterium-environment interaction [22]. Most bacterial phyla possess type IV pili (T4P) [23,24], multi-purpose nanomachines that act via dynamic cycles of extension and retraction mediated by cytoplasmic motor ATPase-driven polymerization and depolymerization of pilin subunits [25–30]. To date, three sub-types of T4P are known of which sub-type 'a' is most prevalent [24]. T4aP are involved in a variety of functions [22]. Among these, surface adhesion, bacterial motility and the uptake of environmental DNA are tightly connected to bacterial virulence [31,32]. It is thus not surprising that several Gram negative and Gram positive human pathogens use T4aP for processes connected to host infection [31,33–38]. In *Acinetobacter*, T4aP play a role in cell adhesion [39], twitching motility and natural transformation [28,40–42]. Moreover, the two-component regulatory system BfmRS, which is important for survival of *A. baumannii* in a murine pneumonia model, also controls T4aP production [43]. While this suggests that T4aP could also drive *A. baumannii* virulence, the most comprehensive comparative genomics study so far between pathogenic and a-pathogenic *Acinetobacter* species failed to detect differences that could hint towards lineage-specific changes in T4aP formation or function [21].

T4aP are prevalent in bacteria irrespective of their lifestyle [22]. Their recurrent recruitment by pathogens for processes connected to host infection therefore suggests that only

considerably subtle modifications are necessary to transform T4aP into a virulence factor. Therefore, any adaptive changes in pathogenic *Acinetobacter* might have escaped the attention thus far, because they require higher resolving analyses beyond determining the presence/absence of T4aP components. Indeed, structural variants of the major pilin subunit PilA were recently detected in *A. baumannii* strains. This suggested the existence of functionally diverse T4aP in this species [39,44]. A comprehensive analysis on this level of resolution for all T4aP components considering, at the same time, the wealth of sequence data covering the full range of *Acinetobacter* diversity is missing. Therefore, it is still unclear to what extent T4aP differ between members of this genus and what consequences this may have for the interaction of *A. baumannii* with the human host.

Here, we increase the resolution of the comparative analysis of T4aP components to the level of individual protein features, such as the presence of Pfam- or SMART domains, of transmembrane domains and of low complexity regions [45]. For each protein, we integrate the annotated features from N- to C-terminus into a feature architecture and compare these between orthologs of the T4aP components. We then exploit that differences in the feature architectures of two orthologs serve as a proxy of their functional divergence [45]. To thoroughly chart the extent of variation in the precise design of 20 T4aP across the genus *Acinetobacter*, we integrated their genus-wide phylogenetic profiles across more than 884 bacterial isolates with an assessment of feature architecture similarity. Three candidates organized in the same gene cluster have altered feature architectures in most pathogenic *Acinetobacter* isolates, which indicates a change in function. A subsequent evolutionary characterization integrated with modelling of their 3D structures and a downstream experimental characterization identifies the pilus tip adhesin ComC as the most prominent candidate driving the functional diversification of T4aP in pathogenic *Acinetobacter*. In summary, our results provide first-time evidence that pathogenic *Acinetobacter* have modulated the precise function of T4aP by changing the structural layout of the pilus tip adhesin.

## Results

### Phylogenetic profiles of Type IV pilus components

*Acinetobacter* T4aP components are best characterized in the naturally transformable bacterium *A. baylyi* ADP1 (Fig 1A), and we used the protein set provided by Averhoff et al. 2021 [46] to prime our analysis. We additionally considered the prepilin peptidase PilD because the corresponding gene is part of the *pilBCD* operon in *A. baylyi*, and because it is likely involved in T4aP biogenesis [40,41]. All *A. baylyi* ADP1 T4aP components are represented in the *A. baumannii* type strain *Ab* ATCC 19606[T] (Table 1), which allowed to reconstruct the evolutionary history of T4aP from the perspective of *A. baumannii*. We first performed a targeted ortholog search for the individual *Ab* ATCC 19606 [T] T4aP components across 855 *Acinetobacter* isolates that cover the diversity of the genus complemented with 29 outgroup species. The resulting presence/absence patterns of orthologs are summarized in the phylogenetic profiles shown in Fig 1B. Orthologs to all T4aP components are found throughout the genus *Acinetobacter*. In the outgroup species, however, the phylogenetic profiles become more sparsely filled. To investigate possible reasons for the non-detection of orthologs in these species, we focused on the gene triple *comB*, *pilX* and *comC*. The three genes reside next to each other in the genome of *Ab* ATCC 19606[T] where they are flanked by *pilV* and *comE*. In *Pseudomonas aeruginosa* PAO1, orthologs were identified only for *pilV* and *comE* (see Fig 1B), but like the situation in Ab ATCC 19606[T], the two corresponding genes flank several T4aP components [47]. Of these, two show a weak albeit significant amino acid sequence similarity to ComB and ComC from *A. baumannii* respectively (S1 Fig). For *Ab* ATCC 19606[T] PilX, we found no

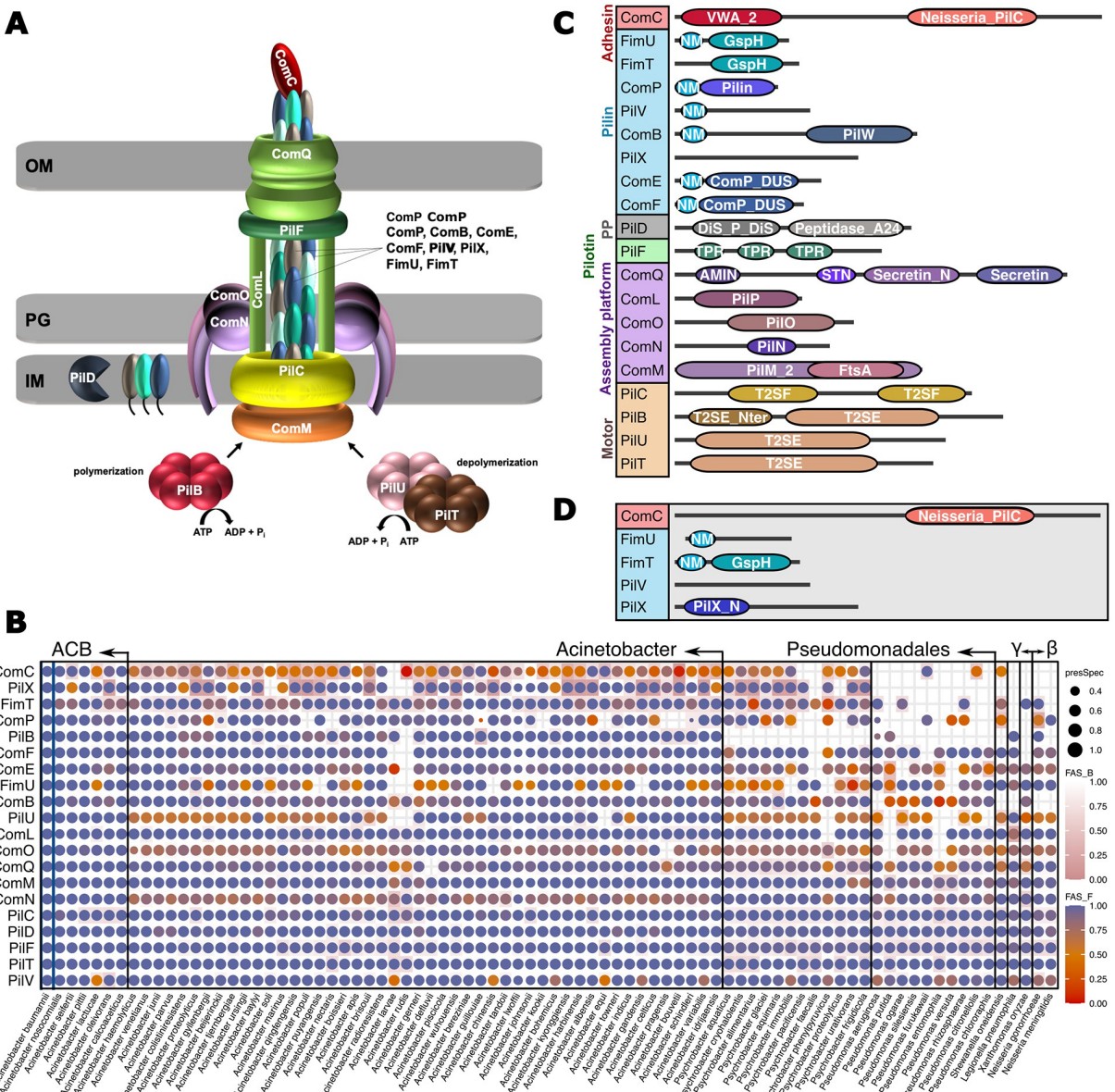

**Fig 1. Characterization of *Acinetobacter* type IVa pilus components.** (A) Model of the type IVa pilus (T4aP) in *A. baylyi*. OM–Outer membrane; IM–Inner membrane; PG–peptidoglycan. The individual components are listed in Table 1. (B) Phylogenetic profiles of the *Ab* ATCC 19606 T T4aP components across 884 bacterial isolates representing 83 named species. Taxa are summarized on the species level. A dot indicates the presence of an ortholog in the respective species, and the dot size represents the fraction of the subsumed isolates an ortholog was identified in. The color encodes the median feature architecture similarity (FAS score) between the protein in *Ab* ATCC 19606 T and its orthologs within a species. The dot color gradient (FAS_F; blue to orange) captures architecture differences using the feature architecture of the *Ab* ATCC 19606T protein as reference. The score decreases if features seen in the *Ab* ATCC 19606T protein are missing in the respective orthologs. The cell color gradient (FAS_B; white to pink) captures architecture differences using the feature architecture of the ortholog as reference. The score decreases if features seen in the ortholog are missing in the *Ab* ATCC 19606T protein. Taxa are ordered according to increasing phylogenetic distance to *A. baumannii* using the information provided in [21] and [100]. 'γ' and 'β' represent γ- and β-*proteobacteria*, respectively. The full data is available as S1 Data. (C) Pfam domain architectures of the T4aP components in *Ab* ATCC 19606 T. Protein lengths are not drawn to scale. Pfam accessions and domain descriptions are available in S1 Table. PP–Prepilin peptidase; NM–N_Methyl. (D) Most common alternative domain architectures for ComC, FimU and FimT represented in (C). Architectures represent ComC, FimU, and PilV in *A. baumannii 1297*, FimT in *A. baylyi*, and PilX in *A. baumannii AYE*.

**Table 1. T4aP components in *A. baylyi* ADP1 and in *A. baumannii* ATCC 19606 [T].**

| Gene name[§] | A. baylyi ADP1 | A. baumannii ATCC 19606 [T] | Locus | Annotation |
|---|---|---|---|---|
| pilU[&] | WP_004922051.1 | WP_000347036.1 | 124856–125974 | PilT/PilU family type 4a pilus ATPase |
| pilT[&] | WP_011182164.1 | WP_000220756.1 | 126002–127069 | type IV pilus twitching motility protein (pilus ATPase) |
| pilF[&] | WP_011182074.1 | WP_000537691.1 | 636841–637608 | type IV pilus biogenesis/stability protein PilW |
| pilB[&] | WP_004920473.1 | WP_001274985.1 | 831958–833670 | type IV-A pilus assembly ATPase PilB |
| pilC[*] | WP_004920476.1 | WP_000279216.1 | 833700–834926 | type II secretion system F family protein |
| pilD[&] | WP_004920478.1 | WP_001152283.1 | 834926–835786 | prepilin peptidase |
| comM/pilM[&] | WP_004923715.1 | WP_000537766.1 | 1511699–1512721 | pilus assembly protein PilM |
| comN/pilN[&] | WP_004923716.1 | WP_000201227.1 | 1512721–1513362 | PilN domain-containing protein |
| comO/pilO[&] | WP_004923719.1 | WP_000076101.1 | 1513359–514099 | type 4a pilus biogenesis protein |
| comL/pilP[&] | WP_004923726.1 | WP_000695065.1 | 1514110–1514637 | pilus assembly protein PilP |
| comQ/pilQ[&] | WP_004923730.1 | WP_001017037.1 | 1514700–1516865 | type IV pilus secretin family protein |
| comP/pilA[&] | WP_004923779.1 | WP_000993713.1 | 1529663–1530091 | pilin |
| fimU[&] | WP_004923829.1 | WP_001214059.1 | 1539389–1539862 | prepilin-type N-terminal cleavage/methylation domain-containing protein |
| pilV[&] | WP_004923832.1 | WP_002194578.1 | 1539856–1540416 | type IV pilus modification protein |
| comB/pilW[$] | WP_004923834.1 | WP_000079195.1 | 1540417–1541418 | PilW family protein |
| pilX[$] | WP_004923837.1 | WP_086221418.1 | 1541475–1542233 | pilus assembly protein |
| comC/pilY1[$] | WP_004923840.1 | WP_085940514.1 | 1542341–1546099 | VWA domain-containing protein |
| comE/pilE[&#] | WP_004923843.1 | WP_001046417.1 | 1546112–1546594 | prepilin-type N-terminal cleavage/methylation domain-containing protein |
| comF/pilE[&#] | WP_004923844.1 | WP_000788344.1 | 1546591–1547016 | prepilin-type N-terminal cleavage/methylation domain-containing protein |
| fimT[&] | WP_004922618.1 | WP_000477156.1 | 1756167–1756682 | type II transport protein GspH |

[§] gene name in *P. aeruginosa* is given after the ‚/' if it differs from the gene name in *A. baylyi*

[*] Pseudogene in *P. aeruginosa*

[&] inferred by orthology

[$] inferred by conserved gene order

[#] *A. baylyi* comE and comF are co-orthologous to *P. aeruginosa* pilE

homolog in *Pa* PAO1 by means of sequence similarity. However, in both species a gene annotated as *pilX* is placed at an identical position in the gene clusters of the two species (see S1 Fig). Integrating all evidences indicates that all three *A. baumannii* genes have orthologs in *P. aeruginosa*, which have been overlooked due to their extensive sequence divergence. Thus, at least part of the gaps in the phylogenetic profiles of T4aP components is due to a limited sensitivity of the ortholog search [48].

## Feature architecture changes in T4aP components

The phylogenetic profiles provide no evidence for a lineage-specific modification of T4aP within *Acinetobacter* that is driven by the gain or loss of individual genes. We therefore increased the resolution of the analysis by comparing the feature architectures of the *Ab* ATCC 19606 [T] T4aP components to those of their orthologs (Fig 1B–1D). For most proteins, feature architectures are either conserved across the genus, or differ only in the presence/absence of low complexity regions or coiled coil regions (see S2 Fig). However, the feature architectures of five proteins, PilX, FimT, PilV, FimU, and ComC deviate to an extent between orthologs that a functional diversification is conceivable [45]. Of these proteins, PilX and FimT are unlikely to drive T4aP diversification on a larger scale. The feature architectures of many PilX orthologs differ from that of the protein in *Ab* ATCC 19606[T] by the absence of a PilX N-term Pfam domain (PF14341; see Fig 1C and 1D). Upon closer inspection we found

that in proteins annotated with PF14341, E-values often only barely meet the detection threshold (S3 Fig). Thus, the sequence variation of *A. baumannii* PilX N-termini appears not sufficiently captured in the Pfam pHMM. This explains why this feature is often missed. The situation is different for FimT. Here, the feature architecture differences are caused by the sporadic non-detection of a N-terminal methylation motif, for example in FimT$_{Ab\ ATCC\ 19606}$ (PF079631; Fig 1D and S4A and S4B Fig). In this protein, the absence of PF079631 from the feature architecture coincides with a threonine at a position in the amino acid sequence that is typically occupied by a leucine or an isoleucine in PF07963 (S4C–S4E Fig). This position is part of a prepilin peptidase cleavage motif ([49]; S5 Fig), and interestingly the change to a threonine is one of the two substitutions that characterizes *A. baumannii* strains that lost the ability to twitch ([10]; S5 Fig). Thus, there is evidence that functionally different variants of FimT exist in *A. baumannii*, however they are rare. More compelling are the findings for PilV, FimU and the pilus tip adhesin ComC. Each of these three proteins exist in two main feature architecture variants in the genus (Fig 1D and S6 Fig). PilV$_{Var1}$ differs from PilV$_{Var2}$ by the annotation of an N-terminal methylation motif (PF07963). FimU$_{Var1}$ differs from FimU$_{Var2}$ by the annotation of a GspH Pfam domain (PF12019), a feature that is seen in pseudopilins which are involved in bacterial type II export systems. Eventually, ComC$_{Var1}$ differs from ComC$_{Var2}$ by the annotation of a Pfam VWA_2 domain (PF13519) in the N-terminal region.

## Evolutionary histories of ComC, FimU and PilV in *Acinetobacter*

Differences in the feature architectures of ComC-, FimU-, and PilV orthologs are first indications for a functional diversification of T4aP components within *Acinetobacter*. We next investigated the evolutionary histories of the three proteins in greater detail. A phylogenetic analysis of the ComC orthologs revealed that the species *A. baumannii* and the genus *Acinetobacter* are both paraphyletic regarding this locus (Fig 2 and S7 Fig). To exclude the possibility that the paraphyly is an artefact of insufficient phylogenetic signal in the data, we confirmed that an alternative tree topology with monophyletic *Acinetobacter* isolates explains the data significantly worse (AU test; p = 0.002; [50]). In the genome of *Ab* ATCC 19606 [T], all three genes (*comC, fimU and pilV)* reside in close vicinity (see Fig 3). To determine whether they are not only physically but also genetically linked, we labeled each taxon in the ComC tree with the respective variant combination for ComC, FimU and PilV. Within *A. baumannii*, but also for most isolates from the *Acinetobacter calcoaceticus-baumannii* (ACB) complex, ComC$_{Var1}$ is almost exclusively found together with FimU$_{Var1}$ and PilV$_{Var1}$. ComC$_{Var2}$ is typically associated with FimU$_{Var2}$ and PilV$_{Var2}$ (Fig 2 and S7 Fig). The association of the variants is only broken up in early branching *Acinetobacter* species outside the ACB complex. Interestingly, the tree reveals a third clade comprising individual *A. baumannii* isolates and one representative of *A. calcoaceticus*. The feature architectures of the three candidate proteins in this clade resemble that of Var1 (S8 Fig), but a higher resolving analysis reveals differences in the case of ComC (see below). To give credit to the distinct phylogenetic placement of this clade, we refer to it as Var1-2 to distinguish it from the more abundant Var1-1.

The phylogenetic analysis has revealed that ComC, FimU and PilV together exist as three evolutionarily distinct lineages with two alternative feature architecture layouts in *A. baumannii*. This suggests that recombination may have affected the evolution of this locus. We next used the change in the pattern of shared variants between the bacterial isolates to assess the length of the genomic region that was likely involved in these recombination events [51]. We included 5 kbp upstream and 3.5 kbp downstream of ComC in the analysis such that the investigated genomic region harbors four additional T4aP components as well as six flanking genes

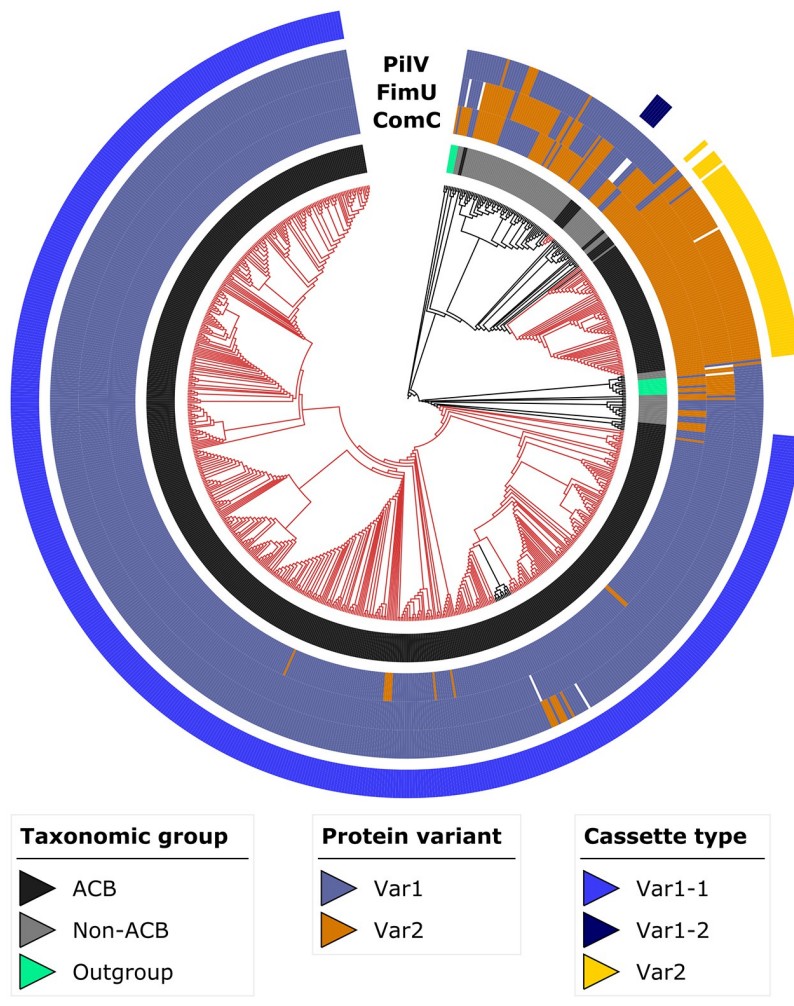

**Taxonomic group**
- ▷ ACB
- ▷ Non-ACB
- ▷ Outgroup

**Protein variant**
- ▷ Var1
- ▷ Var2

**Cassette type**
- ▷ Var1-1
- ▷ Var1-2
- ▷ Var2

**Fig 2. Maximum likelihood phylogeny of ComC.** Red branches indicate *A. baumannii* isolates and black branches isolates from all other species. The inner ring (ring 1) identifies the taxonomic group the isolates in the tree are assigned to. The variant assignments of PilV, FimU and ComC across the investigated taxa are provided in rings 2 to 4, respectively. The three evolutionary cassettes present in the ACB clade (see main text) are represented in the outermost ring 5 (see main text for further details). A high-resolution version of this tree is given in S7 Fig. ACB–*Acinetobacter calcoaceticus-baumannii* complex.

with different functions (Fig 3). To rule out that changes in gene order represent a physical barrier to recombination, we confirmed that the order of these 13 genes is conserved across the *Acinetobacter* diversity (S9 Fig). The analysis revealed that the recombination block spans all seven T4aP components in this region but excludes the flanking genes. This observation integrates well with the finding that the gene tree of the concatenated T4aP components leaves *A. baumannii* paraphyletic (see Fig 3), whereas the gene tree based on a concatenation of the five flanking genes supports monophyletic *A. baumannii* (S10 Fig).

Integrating the results of the evolutionary analyses with the feature architecture variant assignments for ComC, FimU and PilV reveals that the entire cluster of T4aP associated genes represents an evolutionary cassette. The different domain architecture layouts of the three proteins characterize two main variants of this cassette, and an exchange of this cassette occurred at least twice during *A. baumannii* diversification.

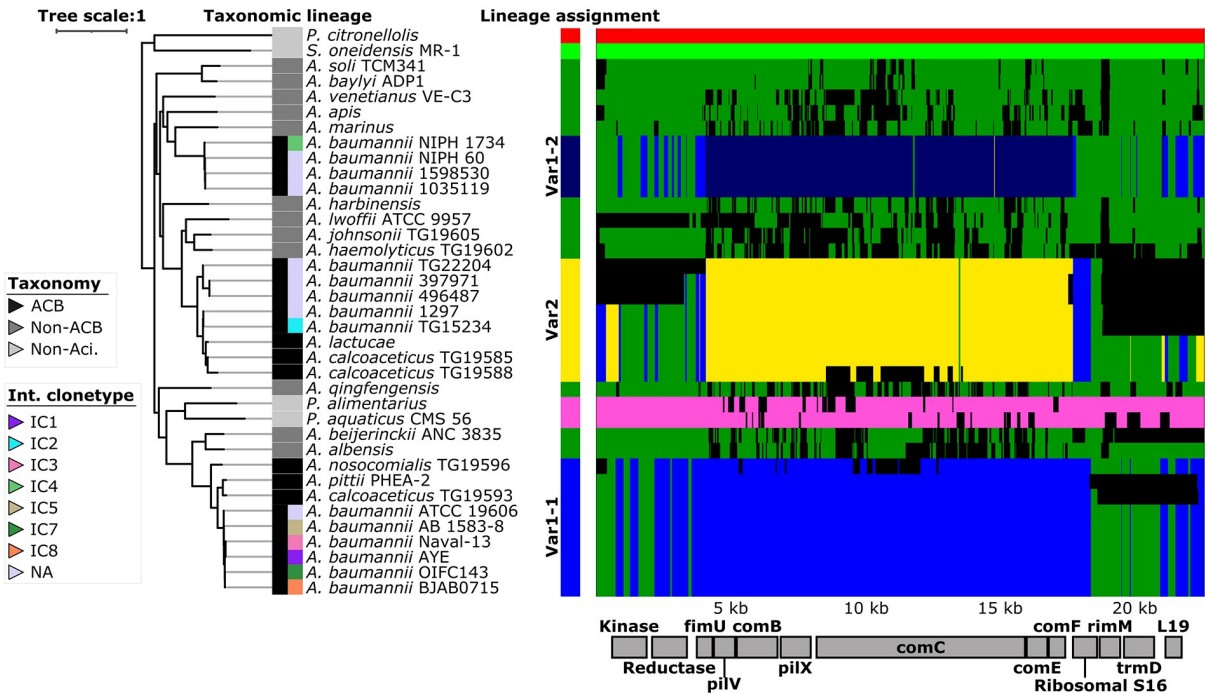

**Fig 3. Patterns of ancestral recombination in the locus surrounding the *comC* gene.** The main plot displays the inferred patterns of ancestral recombination for *Ab* ATCC 19606^T *comC* and the 12 flanking genes across 37 *Acinetobacter* isolates. Identity and order of the genes is indicated by the boxes below the plot. Each of the seven detected genetic lineages is represented by one color, and changes in color along the 13 genes indicate a recombination event. Alignment gaps or sites of unknown source of recombination are shown in black. The three genetic lineages represented in *A. baumannii* isolates are named Var1-1, Var1-2, and Var2 respectively (see main text). The maximum likelihood tree is based on the concatenated alignment of the seven T4aP proteins in this region. Branch lengths represent evolutionary time in substitutions per site. The taxonomic assignments together with the international clonetype (ICs) assignments for the *A. baumannii* isolates [86] are indicated by the colored boxes left of the taxon labels. Multilocus sequence typing of the *A. baumannii* isolates according to the Pasteur scheme is provided in S3 Table and S7 Fig. *P. citronellolis*–*Pseudomonas citronellolis*; *P. alimentarius*–*Psychrobacter alimentarius; P. aquaticus*–*Psychrobacter aquaticus; S. oneidensis*–*Shewanella oneidensis*.

## 3D modelling of ComC reveals variant-specific structural variation

In the highest resolving analysis, we assessed how the differences seen between orthologs of ComC on the feature architecture level are reflected in the predicted 3D structures (Fig 4 and S11 Fig; see S12 and S13 Figs for FimU and PilV, respectively). ComC is characterized by the presence of two globular domains that are connected by a linker (Fig 4A). The C-terminal domain shows considerably little structural variation across the investigated proteins (see Fig 4B panel 3). It comprises the part of the ComC sequence that is consistently annotated with the Neisseria_PilC Pfam (PF05567) domain across all ComC orthologs (see Fig 1C). In contrast, the N-terminal half of ComC is structurally more variable (see Fig 4B panel 2), and this does not coincide with a greater uncertainty in the model accuracy for this part of the predicted 3D structure (see S11 Fig). In ComC_{Var1-1}, the N-terminal part is predicted to fold into an α/β doubly wound open twisted beta sheet conformation, which is surrounded by 7 parallel alpha helices arranged in a cylindrical conformation and an external alpha helix (S14A Fig). This fold agrees with previous structural characterization of the von Willebrand factor type A (vWFa) domain [52,53], and of the vWFa domain in integrin α II b (PDB: 3NIG). A similar structural layout is predicted also for ComC_{Var1-2} (S14B Fig), and both findings are consistent with the annotation of a Pfam VWA_2 domain in the N-terminal part of these variants. However, ComC_{Var2} forms a similar fold (S14 C), and all three variants share the presence of a

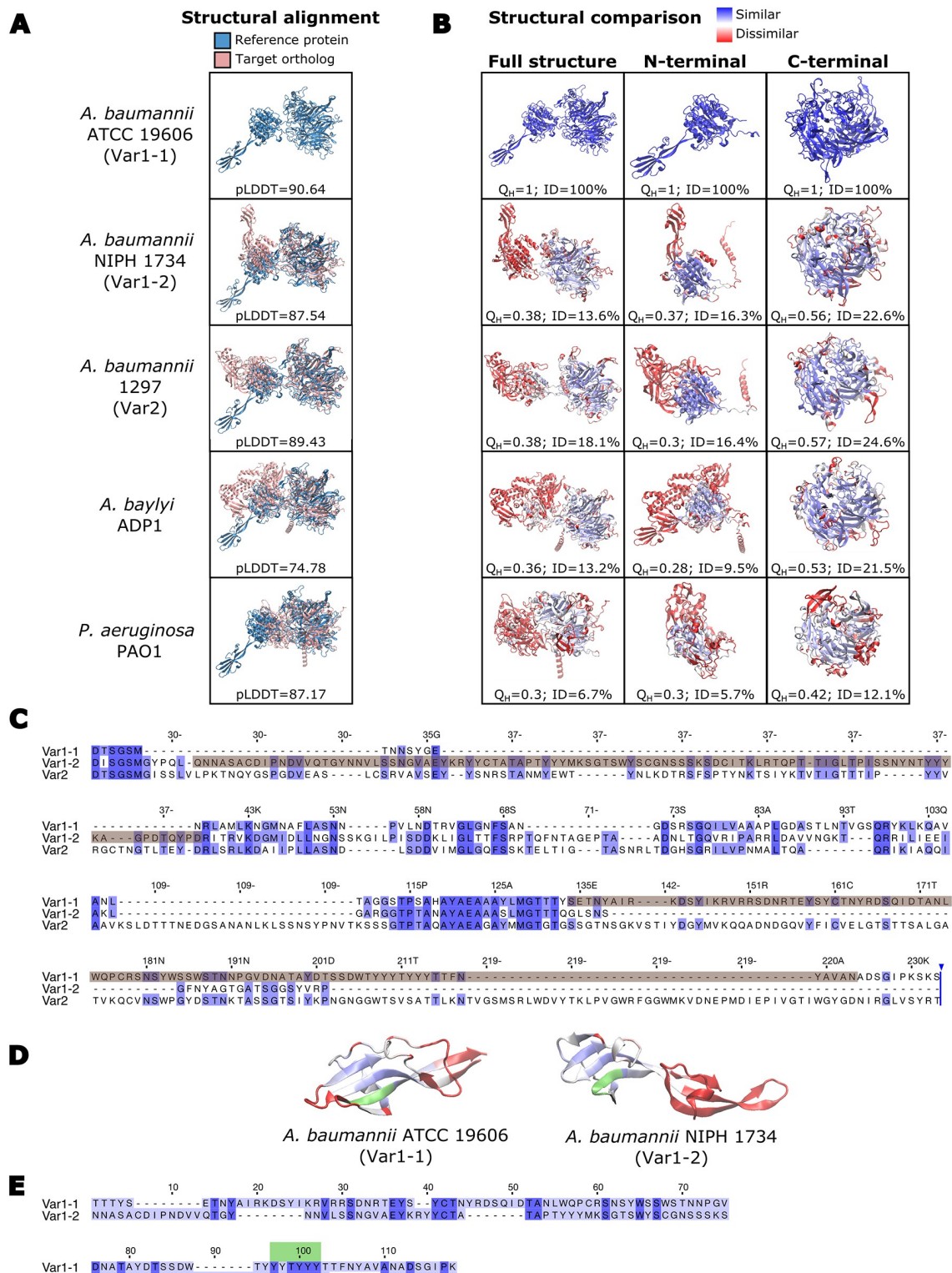

**Fig 4. Structural and sequence variation of ComC.** (A) Alignments of the modelled 3D structures for *Ab* ATCC 19606 [T] ComC (blue) and the ComC of other bacterial isolates (red). The per-residue confidence for each modelled structure is given as the pLDDT value below the structures. Values >90 indicate a high accuracy and values between 70 and 90 a good accuracy of the prediction [114]. (B) Extent of structural conservation between ComC proteins shown in (A) and the reference protein in *Ab* ATCC 19606 [T]. The structure of the target protein is colored with a gradient from blue (high conservation) to red (low conservation). $Q_H$: pair-wise

structural conservation score ranging from 1 (structurally identical) to 0 (no similarity). ID: percent of sequence identity in the structural alignment. (C) Multiple sequence alignment of the N-terminal part of representatives for the three ComC variants in *A. baumannii*. The alignment covers the amino acids 24–233 of *Ab* ATCC 19606 $^\mathrm{T}$ ComC. The sequences forming the finger-like protrusions in ComC$_{Var1-1}$ and ComC$_{Var1-2}$ are shaded in brown. The sequences for the three variants represent the corresponding species shown in (A). (D) Structural similarity between the finger-like protrusions of ComC$_{Var1-1}$ and ComC$_{Var1-2}$. The color gradient from blue to red indicates decreasing similarity. The tyrosine-rich motif is highlighted in green. (E) Pair-wise sequence alignment between the sequences forming the finger-like protrusion in ComC$_{Var1-1}$ and ComCVar$_{1-2}$. Conserved residues are indicated in dark blue; the tyrosine-rich motif is shown in green.

metal ion-dependent adhesion (MIDAS) motif, which is characteristic of vWFa domains [54] (S14 Fig). Along the same lines revealed a VAST analysis [55] that ComC proteins from species across the diversity of the *Acinetobacter* genus display local structural similarities to vWFa containing proteins in the Protein Data Bank (PDB) (S15 Fig). What then characterizes ComC$_{Var1}$ exactly if not the presence of a vWFa domain, as it was initially suggested by the feature architecture comparison (see Fig 1C and 1D)? Fig 4 shows that both ComC$_{Var1-1}$ and ComC$_{Var1-2}$ share the presence of a finger-like protrusion harboring a Tyr-rich motif. Note that these fingers are embedded into the vWFa domains of both variants, however they are differently positioned both in the predicted structures (Fig 4A and 4B) and in the respective amino acid sequences (Fig 4C and S14A and S14B Fig). Therefore, the two fingers are very likely of different evolutionary origins although their structural similarity and the shared presence of the Tyr-rich motif suggest that they originated from the same source (Fig 4D and 4E). Neither ComC$_{Var2}$ in *A. baumannii* or in *A. baylyi*, nor PilY1 in *P. aeruginosa* are in possession of a similar protrusion (Fig 4A and 4B). However, the VAST analysis revealed that these proteins carry other insertions in the region that likely forms a vWFa domain (see S15 Fig).

## Functional role of ComC in *A. baumannii*

The *in-silico* analysis has provided substantial evidence for a hitherto unknown diversity of T4aP within *A. baumannii*. A prominent driver of this diversity is the pilus tip adhesin ComC, and here specifically the N-terminal region that folds into a vWFa domain. ComC$_{Var1}$ differs from ComC$_{Var2}$ by the presence of a finger-like protrusion that has been integrated into the vWFa domain. Moreover, ComC$_{Var1-1}$ displays a local structural similarity to mechanosensitive β3-integrins [56] (see S15 Fig), which belong to a superfamily of cell adhesion receptors in animals [57]. With the following experiments, we shed initial light on the functional relevance of the vWFa domain variant in ComC$_{Var1-1}$ and of the finger-like protrusion therein. We created three different ComC constructs: the full-length version of *Ab* ATCC 19606$^\mathrm{T}$ *comC*, a truncated version that lacks the subsequence that is annotated with the Pfam VWA_2 domain (*comCΔVWA*), and a version where we exclusively deleted the region that encodes the finger in ComC$_{Var1-1}$ (*comCΔ166–256*). Note that a comparison of the predicted structures for ComC and for the ComCΔ166–256 mutant provided no evidence for a mis-folding of the mutant (S16 Fig). The subsequent experiments were performed in a *comC* knock-out mutant of *Ab* AYE-T, because *Ab* ATCC 19606 $^\mathrm{T}$ did neither twitch nor to take up environmental DNA in our hands.

We initially confirmed that *Ab* AYE-T Δ*comC* strain showed no noticeable piliation defect (Fig 5A). This finding is consistent with the observations that a *comC* deletion has no effect on piliation in *A. baylyi* [58] and in *Neisseria* [59], and we conclude that a deletion of *comC* does not interfere with piliation in any *A. baumannii* isolate. Subsequently, we investigated the role of ComC and of the vWFa domain in host cell adhesion (Fig 5B). Compared to wild-type *Ab* AYE-T, a *comC* knock out mutant (*Ab* AYE-T Δ*comC*) displayed a significantly reduced

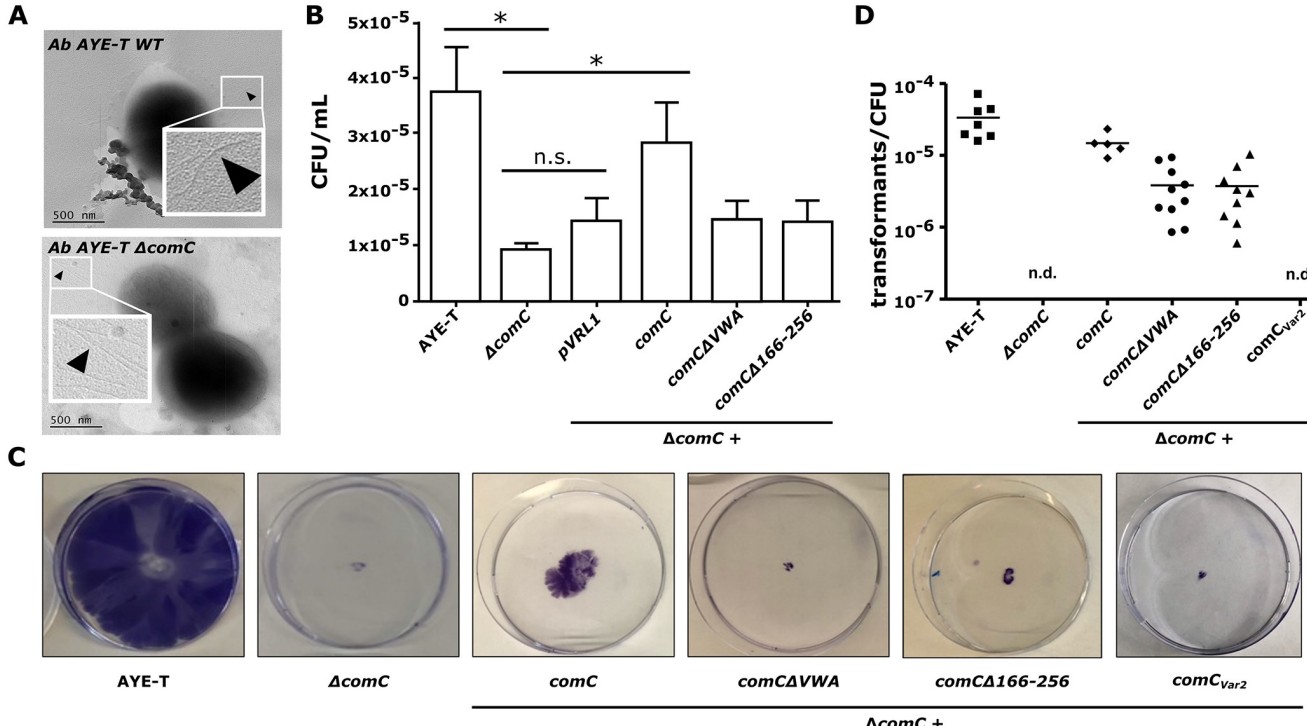

**Fig 5. Functional characterization of ComC in *A. baumannii* AYE-T.** (A) Representative electron micrograph of wild-type *A. baumannii* AYE-T cells (top) and of the Δ*comC* mutant (bottom). No piliation defect is seen for the mutant. A phenotyping of wild-type *A. baumannii* AYE-T, of the Δ*comC* mutant, and of the mutant complemented with the indicated constructs are shown in the following panels. (B) Adhesion to HUVEC cells. The bar height and error bar indicate the mean number and standard deviation, respectively, of colony forming units (CFU) per mL (n = 4; values are given as S2 Table). '*' indicates a significant difference (one tailed t test: p<0.05). Complementation with the truncated *comC* mutants did not significantly increase adhesion rates. (C) Twitching motility. Cells were stained with 1% [w/v] crystal violet. (D) Natural transformation rates of the indicated *A. baumannii* cells were assessed using genomic DNA of rifampicin resistant *A. baumannii* ATCC 19606 T. n.d.—not detected.

adhesion rate to HUVECs (n = 4; t test: p<0.05). Complementing the mutant with the full length *comC* increased the adhesion rates significantly (n = 4; t test: p<0.05), whereas no significant increase was observed when we used the either *comCΔVWA* or *comCΔ166–256* for complementation. We next investigated the role of ComC in T4aP mediated twitching, and for natural transformation. *Ab* AYE-T Δ*comC* showed no twitching motility, and this phenotype was at least partly restored upon complementation with the full length *comC* (Fig 5C). Notably, neither of the truncated *comC* mutants could restore the capability to twitch to a noticeable extent. Like the effect on twitching motility, the deletion of *comC* abolished natural transformation (Fig 5D). Complementation with the full length *comC* almost fully restored the phenotype. Interestingly, this time the complementation with *comCΔVWA* and *comCΔ166–256* also restored natural transformation, however with frequencies that are an order of magnitude below those of the full length *comC*. The partial restoration of natural competence is an independent albeit indirect indication that the truncated ComC mutants do not show a strong piliation defect. To provide further evidence that the observed phenotypes are also not the effect of more subtle pilus instabilities caused by a truncated ComC, we complemented *Ab* AYE-T Δ*comC* with the wild-type ComC_Var2 from *Ab* 17-VT4715T-2. We investigated only the effect of the complementation on natural competence and on twitching, as these phenotypes showed the strongest effect upon *comC* deletion. Notably, ComC_Var2 could neither restore natural competence nor twitching (Fig 5C and 5D).

The experimental data provide first-time evidence that the N-terminal half of ComC$_{Var1-1}$ and the vWFa domain variant contained therein play a critical role in T4aP mediated adhesion to HUVEC cells, for twitching and at least contribute to natural competence. While both twitching and natural competence was abolished upon *comC* deletion, the ability to adhere to HUVEC cells was only reduced. This is best explained by the effect of other adhesins, such as ATA [60], Csu fimbriae [60], or InvL [61], which all contribute to host cell adherence. Interestingly, we observe for all three tested functions the same phenotypic effect when deleting only the finger-like protrusion that is characteristic for ComC$_{Var1-1}$. Thus, that at least part of the ComC$_{Var1-1}$ function seems conveyed by this finger. In line with this hypothesis, we find that ComC$_{Var2}$ cannot rescue a ComC$_{Var1-1}$ knock-out

## Discussion

Type IVa pili are prevalent in bacteria irrespective of their lifestyle where they convey a broad range of functions [22,24]. In some and often only distantly related human pathogens, they represent key virulence factors [32,33,62,63]. This indicates that the precise functions of T4aP have changed multiple times during evolution and probably as an adaptation to differing habitats and lifestyles. The genus *Acinetobacter* harbors environmental bacteria, bacteria that colonize various animals, as well as human pathogens [64]. This diversity in lifestyles provides an optimal setup for tracing also subtle genetic changes that underlie the functional diversification of T4aP that are used by the bacteria to interact with their environment.

The individual components necessary for building up the T4P machinery are almost ubiquitously present across the *Acinetobacter* diversity. This indicates that missing even one factor most likely renders the entire pilus dysfunctional. Along the same lines, it suggests that T4aP are essential for *Acinetobacter* fitness independent of both habitat and lifestyle. However, conspicuous differences between individual T4aP components across *Acinetobacter* isolates emerged on the level of their feature architectures. The connection between feature architecture of a protein and its function is well documented (e.g., [45,65–68]. Therefore, the differences for PilV, FimU and, more prominently for the pilus tip adhesin ComC between *Acinetobacter* isolates point towards a lineage-specific modification of T4aP function.

T4a pilus tip adhesins (T4a-PTA) have received considerable attention in various bacterial species, among them several human pathogens [59,69–71]. Thus far, all investigated proteins share the presence of a Ca$^{2+}$ binding domain in the C-terminal half (Pfam: Neisseria_PilC; PF05567) and assume similar roles in basal pilus function [59,69,72–75]. However, the precise functions of the N- and C-terminal globular domains differ among species. In *Neisseria*, the N-terminal half of PilC1 mediates host cell adherence [59,69,73,74]. In *Legionella*, it is necessary for host cell invasion [76], and in *P. aeruginosa*, it has been associated with the mediation of surface adhesion, mechanosensing, and regulation of pilus retraction [36,71]. In both *Legionella* and *P. aeruginosa*, the host cell adhesion function is mediated by the C-terminal half of the protein [76,77].

Here, we have shown that the most prominent differences between the individual ComC orthologs both within *A. baumannii* and across the genus reside in the N-terminal part. ComC proteins of most *A. baumannii* isolates together with that of some members of the ACB could be exclusively annotated with the VWA_2 Pfam domain. This indicates the presence of a von Willebrand Factor A domain (vWFa) [78], a domain that mediates cell adhesion and cell migration in eukaryotic proteins in [79]. A vWFa domains in a bacterial PTA was first described for the Pi-2a pilus in the Gram-positive *Streptococcus agalactiae*, a leading cause of sepsis and meningitis. As hypothesized from its function in eukaryotes, this domain indeed mediates host cell adhesion [80]. Subsequently, a vWFa domain was also found in PilY1 of *P*.

*aeruginosa* [36,70,81], and supporting its role in bacterial virulence, it was subsequently found that vWFa domains can activate macrophages, central regulators of airway inflammation [82,83]. On the first sight, our findings that exclusively ComC of pathogenic Acinetobacter could be annotated with a Pfam VWA_2 domain suggest that the acquisition of a vWFa domain drives the conversion of T4aP into a virulence factor.

However, the situation appears more complex. Higher-resolving analyses indicate that also ComC of a-pathogenic *Acinetobacter* isolates possess a vWFa domain although the corresponding amino acid sequence is not similar above threshold to the VWA_2 Pfam domain. Evidences include the presence of the characteristic metal ion-dependent adhesion (MIDAS) motif (see S14 Fig; [54]) as well as extended stretches of local structural similarity between the N-terminal region of ComC and eukaryotic vWFa containing proteins (see S15 Fig). What however differentiates the investigated ComC variants are independent insertions into the conserved structural scaffold formed by a vWFa domain (see S14 and S15 Figs). In ComC$_{Var1}$ such an insertion resulted in the formation of a finger-like protrusion (see Fig 4 and S15 Fig). Thus, rather than the presence of a vWFa domain in the PTA it might be the variant of the vWFa domain that determines whether T4aP are involved in virulence related functions, or not.

Testing the function of ComC$_{Var1-1}$ *in-vivo* revealed that this protein is involved in host cell adhesion, twitching and DNA uptake, and that these functions are conveyed by the N-terminal half and of the vWFa domain therein. Interestingly, a deletion of the finger-like protrusion in ComC$_{Var1-1}$ was sufficient for impairing all three processes to an extent that is comparable to the deletion of the entire vWFa domain. Because structural modelling revealed no indication that the deletion of the finger results in misfolding of ComC (see S16 Fig), these findings suggest a functional role of this structure. The observation that wild-type ComC$_{Var2}$ could not complement the deletion of ComC$_{Var1-1}$ further supports this view. Still, our evidence is only preliminary, and further analyses will be necessary to prove the involvement of this finger in ComC$_{Var1-1}$ function. It will then also be interesting to see whether ComC$_{Var1-2}$ differs in function from ComC$_{Var1-1}$, and whether the Tyr-rich motif, that is present in the fingers of both variants has a functional role. Tyrosine assumes a broad spectrum of functions in natural systems, and short tyrosine rich peptides display a high propensity for self-assembly [84]. A functional role of this motif in ComC$_{Var1}$ is therefore conceivable.

Next to ComC, also FimU and PilV vary both in their feature architecture and in the inferred 3D structure across the investigated isolates. The respective variants are tightly associated with the two main ComC variants forming different layouts of the T4aP cassette. The exchange of this cassette, which happened at least twice during *A. baumannii* diversification, likely resulted in structurally, and most likely also functionally different T4aP that may have helped the bacterium to adapt to their specific environment.

The presence of structural variants of PilA (ComP in this study) has previously suggested a functional variation of T4aP in *Acinetobacter baumannii* [39]. Our data did not allow to reproduce this observation, because the structural variation among PilA proteins is not reflected in differences of their feature architectures (see Fig 1). However, reconciling the phylogenetic distribution of the PilA variants with our results reveals discrepant patterns. For example, while Ab ATCC 19606 [T] and Ab ACICU differ in their PilA structure [39], they harbor the same T4aP cassette (see Fig 2). This strongly suggests that the evolutionary and functional plasticity of T4aP is substantially higher than anticipated. Thus, an essential part of how pathogenic *Acinetobacter* isolates interact with their environment, and more specifically with their host, is largely uncharted. It will require highly resolved structural and functional studies to link the various T4aP layouts with lineage-specific differences in T4aP function and to assess the consequences for the bacterial phenotype.

## Conclusion

A broad range of bacterial taxa use Type IVa pili for the interaction with their environment, and their functional diversity has earned them the attribute "Swiss Army knife" among bacterial pili [32]. Here, we provided evidence that T4aP are substantially more diverse in *Acinetobacter* than was hitherto appreciated. Already different isolates within *A. baumannii* seem to differ in their precise T4aP function, and thus in the way they interact with the human host. This rapid change of pilus function seems to be achieved by an evolutionary concept that resembles an interchangeable tool system where the same handle can convey multiple functions depending on the precise layout of the tool head, here represented by the pilus tip adhesin ComC. Increasing the resolution to trace the functional modification of ComC on the subdomain level reveals the same concept. A conserved structural scaffold formed by the von Willebrand factor A domain appears to be structurally and functionally modified by individual and lineage-specific insertions. On a broader scale, our findings suggest that a substantial extent of functional differences between bacteria isolates that is conveyed by changes on the domain- or sub-domain level rather than by the differential presence/absence of genes remains to be detected. Future comparative genomics approaches that aim to unravel the genetic specifics of pathogens should therefore best extend across different scales of resolution. The goal is to integrate lineage-specific differences on the level of gene clusters, genes, protein feature architectures and 3D structural models into a comprehensive reconstruction of molecular evolution in a functional context.

## Materials and methods

### Data collection

Genome assemblies for 855 isolates from the genus *Acinetobacter* were retrieved from the RefSeq database release 204 [85] (last accessed March 07, 2021). International clonetype assignments [86] were adopted from [21]. Multilocus sequence typing was performed with MLSTcheck v2.1.17 [87] using the Pasteur scheme [88]. 27 isolates of closely related γ-proteobacteria and two representatives from the genus *Neisseria* were added as outgroups. The full taxon list is provided as S3 Table.

### Annotation of protein domains

Pfam and SMART domains were annotated with *hmmscan* from the HMMER package v.3.3.2 [89] with the options -E 0.001--domE 0.1--noali using Pfam release 32.0 [90] and SMART release 9 [91]; we further annotated signal peptides with SignalP [92], transmembrane domains with tmhmm [93], coiled-coils with COILS2 [94], amino acid compositional bias with fLPS [95] and low complexity regions with SEG [96].

### Feature-architecture-aware ortholog search

Targeted ortholog searches were performed with fDOG v0.0.8 (www.github.com/BIONF/fDOG) [97] an extended version of HaMStR [98] using *A. baumannii* ATCC 19606 [T] as the reference taxon. Feature architecture similarities between orthologs were computed with FAS v1.3.2 (https://github.com/BIONF/FAS) [68] that is integrated into the fDOG package. Phylogenetic profiles were visualized and analyzed using PhyloProfile v1.8.0 [99]. The taxa represented in the phylogenetic profiles were ordered with increasing phylogenetic distances from *A. baumannii* ATCC 19606 [T] using the *Acinetobacter* phylogeny in [21] and the γ-proteobacteria tree from [100].

## Phylogenetic analyses

Protein sequences were aligned using MAFFT v7.394 with the linsi method [101]. Phylogenetic gene trees were computed using RAxML v8.2.11 [102] with the rapid bootstrapping algorithm and 100 replicates. The WAG model [103] was selected via the PROTGAMMAAUTO option of RAxML as the best-fitting substitution model. Trees were visualized and annotated using the iTOL website [104]. Alternative topology testing was performed using the AU test [50].

## Detection of ancestral recombination

A subset of 33 *Acinetobacter* isolates together with four outgroup taxa were selected to represent both the phylogenetic diversity and the different ComC layouts in our data. For each taxon, we extracted the genomic region between 5,000 bp upstream and 3,500 bp downstream of *comC* and aligned the sequences with MAFFT v7.394. The detection of individual genetic lineages together with the prediction of ancestral recombination events was done with fastGEAR [51] using default parameters. Shared synteny analyses of the genes annotated in the region across the investigated taxa was assessed using the software Vicinator (https://github.com/ba1/Vicinator) [21] based on the orthology assignments from fDOG.

## 3D structure analysis

AlphaFold2 [105] together with the reduced PDB database was used to model the 3D structures for the following proteins: ComC–*Ab* ATCC 19606 [T] (WP_085940514.1), *Ab* 1297 (WP_024436449.1), *Ab* NIPH 1734 (WP_004745407.1), *A. baylyi* ADP1 (WP_004923840.1), *A. qingfengensis* (WP_070071002.1); PilY1 –*Pseudomonas aeruginosa* PAO1 (NP_253244.1) PilY–*Legionella pneumophila* (WP_229293926.1); PilC–*Neisseria gonorrhoeae*. The structures for *Legionella pneumophila* PilY and *Neisseria gonorrhoeae* PilC were retrieved from existing predictions in UniProt (accessions Q5ZXV3 and Q5FAG7, respectively). Models were visualized in VMD [106], and structural conservation between ComC orthologs and *Ab* ATCC 19606 [T] ComC were analysed with the MultiSeq extension from VMD [107] calling the structural alignment tool STAMP [108]. Subdomains and structural similarities to other proteins in the PDB database were identified using the Vector Alignment Search Tool: VAST [55].

## Culture conditions of bacterial strains and cell lines

All *A. baumannii* strains used for experiments in this study are listed in Table 2. Strains were grown in Luria-Bertani medium (LB) at 37°C with 50 μg ml$^{-1}$ kanamycin or 100 μg ml$^{-1}$ gentamicin when needed. Human umbilical vein endothelial cells (HUVECs) were extracted from fresh cord veins and cultivated in endothelial growth medium. Medium was supplemented with growth factor mix and 10% fetal calf serum.

## Generation of mutants and complementation analysis

The *ΔcomC::kanR A. baumannii* AYE-T deletion mutant was generated as described by Godeux, Svedholm (111) using primers 1–10 (S4 Table). All mutants were verified by sequencing. Three different constructs were used to complement *Ab* AYE-T *ΔcomC::kanR*: (i) full length ComC–The *comC* gene plus 700 bp of the upstream region were amplified from chromosomal DNA of *A. baumannii* ATCC 19606[T] using primers 11–12 (S4 Table). The amplicon was integrated into pVRL1 [112] using KpnI and XhoI resulting in plasmid pVRL1_*comC*-19606. (ii) ComC lacking the vWFa domain: pVRL1_*comCΔVWA*-19606 was amplified from pVRL1_*comC*-19606 using primers 13–14 followed by blunt end ligation. (iii) ComC lacking

**Table 2. Bacterial strains used in this study.**

| A. baumannii strains | Genotype or description | Reference |
|---|---|---|
| ATCC 19606[T] | wild type | [109] |
| AYE-T | Deletion of AbaR1 from AYE and repaired comM | [110] |
| AYE-T pVRL1 | With vector pVRL1, gen[R] | This study |
| AYE-T ΔcomC::kanR | AYE-T with comC replaced with kanR, kan[R] | This study |
| AYE-T ΔcomC::kanR pVRL1_comC-19606 | AYE-T with comC replaced with kanR, complemented with comC from ATCC 19606 [T], gen[R], kan[R] | This study |
| AYE-T ΔcomC::kanR pVRL1_comCΔVWA-19606 | AYE-T with comC replaced with kanR, complemented with comC from ATCC 19606 [T] lacking the vWFa domain, gen[R], kan[R] | This study |
| AYE-T ΔcomC::kanR pVRL1_comCΔ166-256-19606 | AYE-T with comC replaced with kanR, complemented with comC from ATCC 19606 [T] lacking amino acids 166–256, gen[R], kan[R] | This study |
| AYE-T ΔcomC::kanR pVRL1 | AYE-T with comC replaced with kanR with vector pVRL1, gen[R], kan[R] | This study |
| 17-VT4715T-2 | wild type | [111] |
| AYE-T ΔcomC::kanR pVRL1_comC$_{var2}$ | AYE-T with comC replaced with kanR, complemented with comC from 17-VT4715T-2, gen[R], kan[R] | This study |

Kan—kanamycin; gen–gentamicin

the finger-like protrusion: amino acids 166–256 were deleted using plasmid pVRL1_comC-19606 and primers 15–16 resulting in plasmid pVRL1_comCΔ166-256-19606. Plasmids were transferred into the mutant *via* electroporation. (iv) *comC* from *A. baumannii* 17-VT4715T-2 was amplified from chromosomal DNA using primers 17/18 and integrated into plasmid pVRL1_comC-19606 which was amplified using primers 19/20. In this way, only the *comC*-19606 gene but not its upstream region was replaced with *comC*-17-VT4715T-2.

## Natural transformation

Wild type and mutant strains were grown in LB medium overnight at 37°C and diluted to $OD_{600}$ 0.01 with phosphate buffered saline (PBS). Equal amounts of the bacterial suspension and DNA (100 ng/μl genomic DNA of rifampicin resistant *A. baumannii* ATCC 19606 [T]) were mixed and 2.5 μl of the mixture were applied onto 1 ml of freshly prepared transformation medium (5 g/L tryptone, 2.5 g/L NaCl, 2% [w/v] agarose) in 2 ml reaction tubes. After incubation for 18 hours at 37°C, cells were resuspended from the medium with PBS. Transformants were selected by plating on selective agar (rifampicin 20 μg ml$^{-1}$).

## Twitching motility

Twitching medium (5 g/L tryptone, 2.5 g/L NaCl, 0.5% agarose) was inoculated by stabbing one bacterial colony through the agar to the bottom of the petri dish. Plates were sealed with parafilm to prevent desiccation and incubated at 37°C for three days. To visualize the cells at the bottom of the petri dish, the agar layer was removed, and cells were stained with 1% [w/v] crystal violet.

## Piliation analyses by electron microscopy

To analyze the piliation phenotype, cells were grown on LB agar plates at 37°C overnight. Cells were prepared for electron microscopy and visualized as previously described [113].

Shadowing of the cells was carried out in an angle of 20˚ (unidirectional) and with a thickness of 2 nm platinum/carbon.

### Analysis of bacterial adhesion to human endothelial cells

Primary human umbilical cord vein cells (HUVECs) were cultivated in endothelial growth medium supplemented with growth factor mix (ECGM, Promocell) and 10% fetal calf serum (FCS, Sigma-Aldrich) in collagenized 75 $cm^2$ cell culture flasks using a humidified incubator with a 5% CO2 atmosphere at 37˚C. HUVECs were seeded into six-well plates and infected with *A. baumannii* (MOI 50) for three hours. The supernatant was removed and cells with adherent bacteria were washed with PBS and detached using a cell scraper. Thereafter, adherent bacteria were quantified by plating serial dilution series. Visualisation and quantification of bacterial adhesion to human endothelial cells was done by fluorescence microscopy as described in [60].

## Supporting information

**S1 Fig. Co-linearity of genes encoding T4aP components in *A. baumannii* and *P. aeruginosa*.** The individual genes are represented by arrows, where the arrow direction indicates the direction of transcription. Gene names are indicated within the boxes. The naming of *A. baumannii* genes follows [46] and the naming of *P. aeruginosa* genes follows [47]. Genes in the two species that are connected by two lines have been identified as orthologs (see Fig 1B in the main manuscript). Genes connected by a single line represent *A. baumannii* proteins that identify the *P. aeruginosa* protein as a best BlastP hit. The BlastP hit E-value is given next to the connecting line. Genes without a connection lack a significantly similar sequence in the respective other species. From conservation of gene order follows that *comB–pilW*, *pilX–pilX*, and *comC–pilY1* are the corresponding orthologs in the two species. NCBI accession numbers: *A. baumannii* ATCC 19606[T]: *pilV*—WP_002194578.1; *comB*—WP_000079195.1; *pilX*—WP_086221418.1; *comC*—WP_085940514.1; *comE*—WP_001046417.1. *P. aeruginosa* PAO1: *pilV*—NP_253241.1; *pilW*—NP_253242.1; *pilX*—NP_253243.1; *pilY1*—NP_253244.1; *pilY2*—NP_253245.1; *pilE*—NP_253246.1.
(PNG)

**S2 Fig. Feature architecture comparisons of *Ab* ATCC 19606[T] T4aP components whose differences between orthologs do not include Pfam or SMART domains.** For the three represented components (A-C), the differences in domain architectures between the protein in *Ab* ATCC 19606[T] and their orthologs in other isolates are limited to coiled coil regions, low complexity regions (LCR) and regions with a compositional bias (FLPS) (see main text, Fig 1). Reference proteins shown in the first column are compared against their orthologs in *Acinetobacter soli* CIP 110264 (GCF_000368705.1). TMM–Transmembrane domain; Gln–Glutamine; Ile–Isoleucine.
(PNG)

**S3 Fig. E-value distribution of PilX_N Pfam domain annotations in PilX orthologs.** Only a subset of PilX orthologs was annotated with a PilX_N Pfam domain (PF14341) whereas the feature architectures of the remaining orthologs comprise only a transmembrane domain (TMM) and a low complexity region (LCR) (A). The box plot shows the E-value distribution for the annotated PilX_N Pfam domains. Most values are high and close to the inclusion threshold of 0.001.
(PNG)

**S4 Fig. Characterization of the pilin FimT.** (A) Pair-wise feature architecture similarities between *Ab* ATCC 19606[T] FimT and its 864 orthologs across the analyzed taxa [68]. The upper block indicates the feature architecture similarity (FAS) scores penalizing the non-detection of an *Ab* ATCC 19606[T] feature in the ortholog (FAS_F). The consistent coloring in blue indicates that the overall domain architecture similarity is high. The lower block penalizes the absence of a feature that is seen in the ortholog but that is absent in *Ab* ATCC 19606[T] FimT (FAS_B). Most orthologs are colored in pink, which indicates that they possess a domain that is not annotated in *Ab* ATCC 19606 [T] FimT. (B) Pairwise feature architecture comparison between FimT in *Ab* ATCC 19606 [T] (WP_000477156.1) and its ortholog in *Ab* AYE (WP_000477147.1). This reveals that *Ab* ATCC 19606 [T] FimT could not be annotated with the N-terminal methylation motif provided by Pfam (PF07963; purple). (C) Pairwise sequence alignment between FimT in *Ab* ATCC 19606 [T] and in *Ab* AYE. Conserved residues are highlighted in dark blue; substitutions are shown in light blue. FimT of *Ab* ATCC 19606 [T] features a threonine at position 9 in the alignment. (D) The HMM weblogo of the N_methyl domain reveals that position 9 is typically occupied by either leucine or isoleucine. (E) Emission probabilities for the 20 amino acids at position 9 in the pHMM representing the Pfam N_methyl motif (PF07963). The emission probability for threonine is 0 at this position, which explains why no N_methyl domain was annotated in FimT of *Ab* ATCC 19606 [T]. ACB–*Acinetobacter calcoaceticus-baumannii* complex; γ–*γ-proteobacteria*; β–*β-proteobacteria*.
(PNG)

**S5 Fig. Comparison of FimT between twitching and non-twitching *A. baumannii* strains.** Twitching and non-twitching *A. baumannii* strains were taken from [10] and the corresponding FimT proteins were aligned. The multiple sequence alignment shows only the parts of the FimT alignment that differ between the investigated strains. The color code represents the degree of sequence conservation. The five non-twitching strains are characterized by two substitutions in FimT. Two strains, including Ab ATCC 19606[T], display an I->T substitution at position 9 in the alignment. This position is part of a prepilin peptidase cleavage motif [49] indicated in green above the alignment. The consensus motif together with the corresponding part of the N-methyl Pfam pHMM is shown on top of the green box (see also S4 Fig). The I->T substitution likely alters the function of the cleavage motif such that it interferes with the processing of FimT. As a consequence, the ability to twitch is lost. Three further non-twitching strains show the canonical prepilin peptidase cleavage motif, however they share an R->H substitution at Pos. 91 in the alignment. Because this is the only difference of these strains to the twitching *A. baumannii* strains, it is tempting to speculate that also the R->H substitution is sufficient to abolish twitching.
(PNG)

**S6 Fig. Domain architecture changes of ComC, FimU and PilV within the genus Acinetobacter.** (A) The main domain architecture (DA) variants of ComC, FimU, and PilV respectively, in the genus *Acinetobacter*. (B) Feature architecture similarity (FAS) scores between each of the three *Ab* ATCC 19606[T] proteins shown in (A) and their orthologs. The protein of *Ab* ATCC 19606[T] is used as reference. FAS scores range from a maximum of 1 (identical architectures) to 0 (no shared domains) [45]. FAS score distributions are given separately for isolates inside (red) and outside (blue) of the ACB complex. The asterisk marks the FAS score mean. The average number of instances per protein (IPP) for the individual domains is represented by the bar plot. Pfam domains: VWA_2—PF13519; Neisseria_Pil_C—PF05567; N_Methyl—PF07963; GspH—PF12019.
(PNG)

**S7 Fig. Maximum likelihood phylogeny of ComC.** The evolutionary relationships of the ComC orthologs are reflected in the left cladogram. *A. baumannii* isolates are indicated by red branches, isolates of all other species are indicated by black branches. The tree including branch lengths is provided to the right of the figure. Annotations show from left to right for each *Acinetobacter* isolate the taxonomic group assignment, the international clonetype, and the strain type according to the Pasteur scheme. Sequence types according to the Pasteur scheme are specified only when more than 10 isolates share the same strain types. Otherwise, they are summarized under 'other'. The next annotation columns provide for each bacterial isolate the protein variants of ComC, FimU and PilV, and the cassette type. A high-resolution version of this tree is available via figshare: https://figshare.com/articles/figure/_/21967694. (PNG)

**S8 Fig. Domain architecture diversity for ComC, FimU and PilV in the genus *Acinetobacter*.** The tree shows a selection of taxa from the full set that represents the diversity of domain architectures of ComC, FimU, and PilV. The color code of the leaf labels resembles that of Fig 3 in the main text. The Pfam domain architectures of the three proteins in the represented isolates are given next to the taxon names. (PNG)

**S9 Fig. Gene order conservation analysis in the genomic region harboring ComC and FimU.** Each line in the plot represents the bacterial isolate that is indicated to the left. The color coding of the taxon labels corresponds to their lineage assignment in Fig 3 of the main text. Each box represents an ortholog to one of the 13 genes in the ComC/FimU region of *Ab* ATCC 19606$^{T}$ (see main Fig 3) and the gene identity is given by the box label. The order of boxes corresponds to the order of genes in the genome of the respective isolate. *comC* orthologs are marked in white. Green and red boxes identify genes that are upstream and downstream of *comC* in *Ab* ATCC 19606$^{T}$, respectively. The number of genes separating any two of the boxed genes or that are placed upstream or downstream up to the scaffold end (indicated by yellow bars) are given in parenthesis. The plot confirms that the gene order in this region is conserved throughout the genus with very few exceptions. (PNG)

**S10 Fig. Maximum likelihood phylogeny of the concatenated alignment of the genes flanking the T4aP components.** The gene tree is based on a multiple amino acid sequence alignment of the genes flanking the T4aP components in Fig 3 of the main manuscript. It comprises the following proteins: guanylate kinase (WP_000015937.1), 4-hydroxy-3-methylbut-2-enyl diphosphate reductase (WP_000407064.1), 30S ribosomal protein S16 (WP_000260334.1), ribosome maturation factor RimM (WP_000189236.1), tRNA (guanosine (37)-N1)-methyltransferase TrmD (WP_000464598.1), and 50S ribosomal protein L19 (WP_000014562.1). NCBI accession numbers are given in parenthesis. Branch labels denote percent bootstrap support. (PNG)

**S11 Fig. Structural models of ComC of the indicated bacterial isolates.** The color coding in the structures represents the per-residue confidence scores (pLDDT) provided by AlphaFold2 [114]. All structures are arranged such that the N-terminal domain is oriented to the left and the C-terminal domain to the right. (PNG)

**S12 Fig. Comparison of two PilV variants.** (A) Comparison of the modelled 3D structures of PilV$_{Var1}$ (*A. baumannii* ATCC 19606T) and PilV$_{Var2}$ (*A. baumannii* 1297). The structures are

colored according to their mutual structural similarity from high (blue) to low (red). (B) Pairwise sequence alignment of the two PilV variants. Residue numbering refers to PilV in *Ab* ATCC 19606$^T$. The non-detection of the N-terminal methylation motif (PF07963) in PilV$_{Var2}$ is due to the substitutions at positions 14 (V->A) and 17 (L->M) in the alignment.
(PNG)

**S13 Fig. Comparison of two FimU variants.** (A) Comparison of the modelled 3D structures of FimU$_{Var1}$ (*A. baumannii* ATCC 19606$^T$) and FimU$_{Var2}$ (*A. baumannii* 1297). The structures are colored according to their mutual structural similarity from high (blue) to low (red). Red colored alpha helices in both structures correspond to the signal peptide. (B) Pairwise sequence alignment of the two FimU variants. Residue numbering refers to FimU in *Ab* ATCC 19606.
(PNG)

**S14 Fig. Structural fold of the N-terminal domain of the ComC orthologs.** Alpha helices (orange) and beta strands (blue) are shown in the 3D cartoon representation (left) and in the secondary structure topology plots of sample proteins (right). Highlighted are the MIDAS motif (yellow), tyrosine-rich motif (light green), subdomains (brown and bright blue), and external folds (grey, purple, dark green). *A. baumannii* ATCC 19606$^T$ (A) and *A. baumannii* NIPH 1734 (B) feature an antiparallel beta sheet finger-like protrusion (brown). (C) The structure of *A. baylyi* contains additional folds (dark blue, purple, green, and brown) that are highly disordered and form a planar shield that surrounds the MIDAS motif.
(PNG)

**S15 Fig. VAST searches for regions of structural similarity in the N-terminal half of ComC for selected bacterial isolates.** For each of the analyzed proteins, the following information is given: Top left–linear domain-representation of the ComC N-terminal region, where different domains are indicated by different colors. The location of the domain in the 3D structure model to the right is given by the correspondingly colored region. Bottom—Segments sharing a significant structural similarity to an existing structure in the PDB database are identified by arrows beneath the linear domain representation, and we limited the information to the three highest scoring hits. Information about the hit protein is given to the right. (A) *A. baumannii* ATCC 19606$^T$ (ComC$_{Var1-1}$); (B) *A. baumannii* NIPH 1734 (ComC$_{Var1-2}$); (C) *A. baumannii* 1297 (ComC$_{Var2}$); (D) *A. baylyi* ADP1; (E) *A. qingfengensis*; (F) *Pseudomonas aeruginosa*; (G) *Legionella pneumophila*; (H) *Neisseria gonorrhoeae*. The analyses reveal that all investigated ComC orthologs display local structural similarities to eukaryotic vWFa domain containing proteins. However, they all differ by the presence of sequences with weak structural similarities to a diverse set of proteins in PDB. This indicates that a structural backbone of a vWFa domain is modified, in a lineage-specific way, by insertions of different origins.
(PNG)

**S16 Fig. Comparison of the predicted structures of *Ab* ATCC 19606$^T$ ComC and of *Ab* ATCC 19606$^T$ ComCΔ156–266.** The predicted structures of Ab ATCC 19606$^T$ ComC and of the mutant lacking the finger like protrusion are shown in panels (A) and (B), respectively. The confidence in the predicted structure for the mutant lacking the finger-like protrusion is high (pLDDT = 90.06) [114]. The hatched box in (A) indicates the finger in the wild-type protein. A structural alignment of the two N-terminal globular domains is shown in (C), where structurally conserved regions are shown in blue. The pLDDT of the mutant is 90.06.
(PNG)

**S1 Table. Pfam domains annotated in type IVa pilus proteins of *A. baumannii* ATCC 19606.**
(XLSX)

**S2 Table. Results of the adhesion experiments.** Values are given in colony forming units per mL.
(XLSX)

**S3 Table. List of taxa analyzed in this study.**
(XLSX)

**S4 Table. Primers used in this study.**
(XLSX)

**S1 Data. Phylogenetic profile of *Acinetobacter baumannii* type IVa pilus components across 884 bacterial strains.** The corresponding data files are available online via figshare: https://figshare.com/articles/dataset/_/21964535 and can be viewed with PhyloProfile [99].
(ZIP)

**S2 Data. Maximum likelihood phylogeny of ComC in Newick format.**
(NWK)

## Acknowledgments

The authors wish to thank all researchers for making annotated genome sequences available to the public domain. We thank M. Linder from the Max-Planck-Institute of Biophysics, Frankfurt am Main, for the support of electron microscopical analyses, Xavier Charpentier from the Claude Bernard University, Lyon for providing strain *A. baumanii* AYE-T, Gottfried Wilharm for providing strain *A. baumannii* 17-VT4715T-2 and Felix Langschied for helpful discussion and critically reading of the manuscript.

## Author Contributions

**Conceptualization:** Ingo Ebersberger.

**Data curation:** Ruben Iruegas.

**Formal analysis:** Ruben Iruegas, Ingo Ebersberger.

**Funding acquisition:** Stephan Göttig, Beate Averhoff, Ingo Ebersberger.

**Investigation:** Ruben Iruegas, Katharina Pfefferle, Stephan Göttig, Ingo Ebersberger.

**Methodology:** Ruben Iruegas, Stephan Göttig, Beate Averhoff, Ingo Ebersberger.

**Project administration:** Ingo Ebersberger.

**Resources:** Stephan Göttig, Beate Averhoff, Ingo Ebersberger.

**Supervision:** Beate Averhoff, Ingo Ebersberger.

**Validation:** Ingo Ebersberger.

**Visualization:** Ruben Iruegas, Ingo Ebersberger.

**Writing – original draft:** Ruben Iruegas, Ingo Ebersberger.

**Writing – review & editing:** Ruben Iruegas, Katharina Pfefferle, Stephan Göttig, Beate Averhoff, Ingo Ebersberger.

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
