## [Decision Letter · Decision Letter 0]

6 Mar 2023

Dear Dr Ebersberger,

Thank you very much for submitting your Research Article entitled 'Domain-architecture aware phylogenetic profiling indicates a functional diversification of type IVa pili in the nosocomial pathogen Acinetobacter baumannii' to PLOS Genetics.

The manuscript was fully evaluated at the editorial level and by three independent peer reviewers. The reviewers appreciated the attention to an important problem, but raised some substantial concerns about the current manuscript, especially reviewer 3. Based on the reviews, we will not be able to accept this version of the manuscript, but we would be willing to review a revised version that addresses the reviewer's comments. We cannot, of course, promise publication at that time.

If you decide to revise the manuscript for further consideration at PLOS Genetics, please aim to resubmit within the next 60 days, unless it will take extra time to address the concerns of the reviewers, in which case we would appreciate an expected resubmission date by email to plosgenetics@plos.org.

We are sorry that we cannot be more positive about your manuscript at this stage. Please do not hesitate to contact us if you have any concerns or questions.

Yours sincerely,

Xavier Didelot

Academic Editor

PLOS Genetics

Lotte Søgaard-Andersen

Section Editor

PLOS Genetics

Reviewer's Responses to Questions

**Comments to the Authors:**

Reviewer #1: Manuscript PGENETICS-D-23-00119 is an interesting and well written study on the phylogenetic profiling of type IVa pili in Acinetobacter baumannii. The manuscript adds novel information on the mechanisms of adhesion, twitching motility and natural competence in Acinetobacter and is of interest for the readers of the journal. I am listing below specific queries which need to be addressed by Authors.

1. Introduction, lines 66-67: Acinetobacter baumannii is a Gram negative nosocomial pathogen that accounts for about 5% of the total bacterial infections worldwide [1-3]. Authors need to reconsider the statement according to more recent evidence of the literature. I suggest the following three recent publications: Nowak, J. et al. (2017) J. Antimicrob. Chemother. 72, 3277–3282; Wong et al. 2017. Clin Microbiol Rev 30:409–447; Tacconelli et al. 2018. Lancet Infect Dis 18:318–327.

2. Table 1. To avoid confusion, comP and pilA synonyms of pilin should be indicated both in the first column of the table as comP/ pilA instead then in thew footnote.

3. Figure 1B and Figures 2 and 3. Inclusion of Acinetobacter oleivorans into Acinetobacter baumannii-calcoaceticus complex. Authors should indicate the methods/criteria used to include A. oleivorans into A. baumannii-calcoaceticus complex. While it is evident that A. olevoirans DR1 isolates is a distinct species of Acinetobacter genus (Kang et al. J Microbiol. 2011;49(1):29-34), I did not find any A. oleivorans isolate identified at species level using MALDI-TOF (Marí-Almirall et al. Clin Microbiol Infect. 2017; 23: 210.e1-210.e9) or Raman Spectrometry (Molecules 2019, 24, 168). Instead, it is possible to perform species identification of A. oleivorans genomes using ribosomal MLST (Jolley et al. 2012; Microbiology 158:1005-15). A collection of genomes assigned to 72 distinct species into Acinetobacter genus is available at in AcinetobacterPubMLST genomes databases at

https://pubmlst.org/bigsdb?db=pubmlst_abaumannii_isolates&page=query&genomes=1

If identification at species level of Acinetobacter genomes included in the manuscript has been performed using genome sequences and/or ribosomalMLST allele profiles, I suggest to modify genomes classification by grouping A. baumannii, A. nosocomialis, A. pittii, A. seifertii and A. lactucae genomes into A. baumannii group and all other genomes into non-baumannii Acinetobacter group. This approach has been adopted using MALDI-TOF identification for Acinetobacter spp. (Marí-Almirall et al. Clin Microbiol Infect. 2017; 23: 210.e1-210.e9).

4. Fig 3 International clonetypes of A. baumannii genomes. I have a major concern regarding assignment of A. baumannii genomes reported in Fig. 3. It has been shown by several publications that A. baumannii has a clonal population structure dominated by three global/international clonal lineages, named GC1, 2 and 3, and few additional epidemic clonal lineages, which emerged worldwide (Zarrilli et al. Int.J. Antimicro Agents. 2013, 41, 11-19). Among several typing methods used, it has been demonstrated that the most appropriate to study A. baumannii genomes and assign to epidemic clonal lineages is PasteurMLST-based typing scheme (Gaiarsa et al. Front. Microbiol. 10:930, 2019). I invite Authors to assign A. baumannii genomes included in Figure 3 and in Supplementary Table 3 to international clonal types using Pasteur MLST scheme as described in Gaiarsa et al. Front. Microbiol. 10:930, 2019 and taking advantage of additional information found in AcinetobacterPubMLST genomes databases at https://pubmlst.org/bigsdb?db=pubmlst_abaumannii_isolates&page=query&genomes=1

Also, I invite Authors to correct international clonal type assignment of Figure 3. In fact, Acinetobacter baumannii TG215234 and Acinetobacter baumannii ATCC19606 are both assigned to international clonal type 2. This is true for Acinetobacter baumannii TG215234, while A. baumannii ATCC19606 has a distinct clonal type (assigned to PasteurST25 type). Moreover, I noticed that Authors considered in sdupplementary Table 3 an old version of A. baumannii ACICU genome sequence, while thay should consider the revised complete genome sequences of Acinetobacter baumannii strains AB307-0294 and ACICU belonging to global clones 1 and 2 published recently by Hamidian M et al. Insights from. Microb Genom. 2019 Oct;5(10):e000298.

5. Domain architecture difference between FimT orthologs reported in Results section on lines 153-155. Authors should discuss whether this might be responsible for the absence of twitching motility in A. baumannii ACTCC 19606 strains observed in this study and in previous ones.

6. Discussion, lines 350-362.Differences between ACICU and ATCC19606 PilA variants and comC – FimU genes should be explained by differences in genetic background of ACICU and ATCC 19606 genomes.

7. Reference nr. 20. Please, correct title.

Reviewer #2: This manuscript by Iruegas et al describes a pan-genomic comparison of type IV pilus (T4P) genes in the genus Acinetobacter. Based on this comparison, the authors found that for a specific cluster of those genes, recombination events had created substantial variation. In particular, the authors found the insertion of a VWA domain into comC, the tip adhesion, for some strains of A. baumannii, while others have the, apparently ancestral domain structure shared by A. baylyi. Alterations to this domain cause defects in multiple T4P functions in the A. baumannii AYE-T background.

This manuscript is well-written and contains new knowledge for those with an interest in the function and evolutionary origins of T4P systems. Previous analysis of variation in Acinetobacter T4P systems has been focused on the major subunit, pilA, while the fimU/comC variation appears to be the result of a separate, possibly earlier, evolutionary divergence.

The only significant concern I have with the interpretation of the data is how to interpret the loss of T4P function in the comCΔVWA and comCΔ166-256 mutants. While it is possible that the VWA structure plays a role in twitching motility, DNA-uptake and host-adherence, simpler explanation (particularly given the natural occurrence of functional non VWA ComC proteins) is that ComC stabilizes the pilus structure and that the mutants are less stable than wild type ComC, leading to lower ComC expression and fewer pili.

Specific comments:

-Based on figures 1 and 3, it appears that pilV has similar levels of variation and may be covariant with comC and fimU, while only variation in the later two is discussed. Do the pilV genes in Var2, Var1-1 and Var1-2 subsets differ?

-It is no fault of the authors because the naming of T4P genes and gene-products in the literature is a mess, but It may be easier for some readers to follow if the gene names of the Pseudomonas homologues (pilA rather than comP, pilY1 rather than comC) were used, simply because that system has been better studied historically. The authors do include both names in some parts of the text, but some genes (ex. comE and comF are included in Figure 1 but not discussed) A simple table of equivalency could be added to Figure 1 to help readers more familiar with the Pseudomonas (rather than the A. baylyi) T4P nomenclature.

-In Figure 1C, I am concerned that the figure overstates the differences between some of these genes. As the authors correctly point out, the absence of a pfam domain could simply be lack of recognition. I do not believe, for instance, there are any Acinetobacter fimT or pilX genes which lack the N-terminal PilD recognition site or TM helix which make up the N-Methyl domain. Similarly, I would imagine that all Acinetobacter fimU genes are gspH homologues based on the commonly proposed models for the T4Pa tip complex.

-In Figure 4 (panels A and B), it is difficult to follow what is happening with the superimpositions at this figure resolution; however I am concerned that some of the differences are artifacts of attempting to align the whole multi-domain structure. Structure prediction (including AlphaFold) is much better at predicting domain structure than aligning multiple domains and for some (ex. ComC from A. baumannii NIPH 1734 in Figure 4A) the domains may simply be rotated (arbitrarily) the wrong way in the model.

Reviewer #3: This manuscript by Iruegas and colleagues deals with comparative genomics of T4aP systems in Acinetobacter genomes at the protein-domain level, in order to find determinants of pathogenicity in A. baumannii, whenever more classical comparative genomics involving gene repertoires could not find relevant traces. The authors claim to find some specificities in terms of domain compositions for pathogenic strains, and make some experimental testing on the role of these domains in abilities of Acinetobacter to adhere to human cells, to perform twitching motility or competence. The idea is appealing and interesting, as it is true that the information at the domain-level is very often overlooked in comparative genomics, even though it may hold crucial cues to understand bacterial adaptation. However, the manuscript is difficult to read and follow, as the bioinformatic methods in particular, and even the global approach is hard to grasp. In particular, it is unclear how protein domains were annotated (see my comments) and how a “domain loss” (or addition) was inferred, whenever it is crucial for the analysis that look into the evolution of the presence/absence pattern of protein domains. In the end, I am still wondering what really makes the difference between the sequences of the protein-domain variants of FimU and ComC. And can the outcome of a PFAM domain annotation suffice to infer the presence or absence of a functional domain? Considering the potential thresholds effects in such an analysis, I would assume that it should be completed with in-depth multiple sequence alignment analyses.

It is also a bit difficult to understand how the recombination and protein domain analyses results articulate. I finally also wonder how reliable are the structural modelling experiments, and whether the experimental designs proposed to test the functional role of ComC domains highlighted by bioinformatic analyses, are sufficient to support the conclusions, in particular that the authors “provide experimental evidence that this finger conveys virulence-related functions in A. baumannii” (from abstract, see also my comments).

For these reasons, I unfortunately cannot be supportive of the publication of this article. I think the work and approach is valuable and of interest, but that there are missing pieces to make it a convincing case, and an effort to be made on how the results and methods are presented. A synthetic figure on the evolutionary proposed for example could be valuable. Please find more details in my comments below, I hope they will be considered as constructive.

Major comments

------------------

1) Line 131. “Orthologs to all”... Please start the paragraph saying a word on the methodology for "ortholog search", otherwise the entire paragraph is hard to follow. As this is so central to this study’s methodology and results, I believe it should be explained what it consists in, and that providing a link to a Github repository is not sufficient in Methods (Line 396). Are you sure they are orthologs and follow the same gene history? How was it inferred? If not inferred, then maybe could the less precise "homologs" term be used?

2) Line 152-153: rather than “chance effects” that sounds a bit odd, couldn’t the authors rather involve variations in sequences to explain the annotation or not of this domain with PFAM profiles? Depending on their specificity and sensitivity? On line 390 of the Methods, it would be very important to indicate what parameters were used for HMMER? Was the cu-ga option used? Otherwise, which thresholds? And which HMMER version?

We touch here the heart of my main criticism: I do not understand how domains were annotated in details, and how/when a loss of domain was inferred, and how to distinguish it from cases where sequence divergence solely could explain the results.

For example on lines 156-158 the authors then claim themselves “The domain loss in the A. baumannii type strain is caused by a substitution that results in a tyrosine at a position that is typically occupied by a leucine or an isoleucine in PF07963”. Rather than invoking here and elsewhere “domain losses”, could the authors rather involve threshold effects in the annotation? That the domain for example does not pass the threshold?

Again, what is unclear to me here and later on, is whether the domain is really “lost” or gained (and what would mean lost here, it is not an entire domain deletion apparently from figure 1C-D?) or whether the sequence is just divergent from the consensual domain as defined in PFAM. In Discussion (line 314-319), the authors themselves observe: “However, general features of a VWA domain are present also in ComC of a-pathogenic Acinetobacter isolates.” And further ask “Why then were these domains not annotated with the VWA_2 Pfam domain?”

It probably does not change the functional interpretation, this domain diversification might be as well very functionally relevant, but it changes the angle of the overall approach here and the relevance of the underlying bioinformatic analyses.

On lines 166, it is indicated that “We next investigated the evolutionary histories of ComC and FimU in greater detail.” However, I would have expected to see, aside from (or maybe even before) these phylogenetic analyses, an analysis of the multiple sequence alignment of these proteins first, because from the schemes on Fig 1C, it seems that there was not a clear-cut “domain” loss, as the sequences from both variants in both proteins show consistent length. Do they align well? If not, how could it impact the phylogeny? Do the variants clearly differ in sequence at the position of these domains? There is an alignment displayed in Fig4C, but it comes late in the article, and is limited to three sequences, making it difficult to infer how conserved are the different regions of the MSA.

3) Fig. 2 represents a Maximum-Likelihood phylogeny, but neither branch lengths nor branch supports appear. Was there any attempt to root it? These reasons and questionings leave this tree difficult to use for evolutionary interpretations.

4) A recombination analysis followed, but it is difficult to grasp how its results articulate with that of the domain analyses: recombination analyses showed a large tract of recombination spanning several genes (Fig. 3) and yet in Fig S11, it is written in the legend “This indicates that a structural backbone of a VWA domain is modified, in a lineage-specific way, by insertions of different origins.” Could the authors clarify? How many events are invoked for instance?

5) Regarding the recombination analysis, the paragraph should contain a word on which principle was used for recombination detection, to guide the authors through the results and drawn conclusions. Also, since the authors used phylogenetic approaches, could phylogenies of the genes inside/outside the cassette be used to demonstrate different evolutionary histories, hence supporting the cassette and recombination scenario? Could the phylogeny on Fig 3 already be it? I don’t see it much discussed in the main text. Also, the components from Fig S7 could have been used to complement Fig 3.

6) Line 216-218: “Therefore, the two fingers are very likely of different evolutionary origins although their structural similarity and the shared presence of the Tyr-rich motif suggest that they originated from the same source” I found this sentence unclear and convoluted. Please rephrase.

7) I am not a structural biologist, but what is the confidence of the 3D models obtained with AlphaFold in the precise regions of interest for ComC on Fig 4A? I have seen many times these models with mapped confidence scores. Are the features observed to differ between modelled “variants” reliable? What would happen were diverse sequences from a same “variant” chosen for the modelling? What is the confidence in the positioning of the finger-like protruding domain? Or its presence even? Unfortunately, additional information might be present on Fig S11, but the resolution of the image is too low. Could the “finger-like protrusion” region be somehow highlighted or pointed at where relevant on Fig 4?

8) The proposed mutant comC to test experimentally are based on mutant proteins that seem somehow drastic versions of it. Despite the fact that 3D models predict similar folding, again what is the confidence for it?

I am not convinced that the experimental results found in Fig. 5 demonstrate specifically the role of the VWA_2 PFAM (nor or the finger) domain in host cell adhesion, twitching motility or competence. They may demonstrate some role for the presence of the N-ter of the protein for the functioning of the T4aP, but I believe that by design the experiments cannot address the question on how the “content” of the N-ter may affect function. Some exchanges in N-ter experiments between variants, some point mutations of specific aa, or other experimental designs, may further contribute to address this question. For instance, what happens when performing the exact same experiments with corresponding fragments from the variant v2, or v1-2? Would they or not result in similar phenotypes? This should at least be discussed.

Minor comments

------------------

9) Line 36, abstract section: “Both genes together form an evolutionary cassette” is not totally reflecting the scenario proposed by the authors, because the proposed cassette would contain several extra genes form the T4aP, right?

10) I found the end of introduction too detailed on the results, it is somehow redundant with an abstract.

11) I do not see the connection with the paragraph on lines 124-137 and the very 1st line of the paragraph "Acinetobacter T4aP components are best characterized in the naturally transformable bacterium A. baylyi ADP1". Should this sentence be placed elsewhere?

12) Fig. 1B. it is unclear what should be compared here. Also, I do not get the meaning of the FAS_R score? Why do we need two scores here, and what is “the ortholog”? Please clarify to help the reader analyse the figure.

13) Lines 149-150: “the architectures of four proteins, PilX, FimT, FimU, and ComC deviate to an extent between orthologs that it could indicate a change in function.” From Fig. 1B, one could also wonder about the relevance of ComO and ComN.

14) Line 124: Included in what? Please rephrase as it is unclear.

15) Line 170-171, it is mentioned that ComC and FimU are close in the genomes of Ab ATCC 19606 T is this also the case in other Acinetobacter genomes?

16) Figure 2: what are the black branches?

17) Line 185-186: “This indicates that recombination has affected the evolution of this locus” – I am not sure this can be argued at this point of the text. “might have affected” could be more appropriate?

18) Line 386: "RefSeq database release 204" Could the authors mention on what date it was last accessed?

Typos

------

Line 74 : remove «how »

Line324: PDB means “Protein Data Bank”. Authors may want to add in text it is a database of structures of proteins?

Line 413: where => were

**Have all data underlying the figures and results presented in the manuscript been provided?**

Reviewer #1: Yes

Reviewer #2: Yes

Reviewer #3: Yes

PLOS authors have the option to publish the peer review history of their article (what does this mean?). If published, this will include your full peer review and any attached files.

Reviewer #1: **Yes: **Raffaele Zarrilli

Reviewer #2: **Yes: **Kurt Henry Piepenbrink

Reviewer #3: No

---

## [Editor Report · Decision Letter 1]

6 Jun 2023

Dear Dr Ebersberger,

We are pleased to inform you that your manuscript entitled "Feature architecture aware phylogenetic profiling indicates a functional diversification of type IVa pili in the nosocomial pathogen Acinetobacter baumannii" has been editorially accepted for publication in PLOS Genetics. Congratulations!

Yours sincerely,

Xavier Didelot

Academic Editor

PLOS Genetics

Lotte Søgaard-Andersen

Section Editor

PLOS Genetics

**Data Deposition**

http://datadryad.org/submit?journalID=pgenetics&manu=PGENETICS-D-23-00119R1

**Press Queries**

---

## [Editor Report · Acceptance letter]

5 Jul 2023

PGENETICS-D-23-00119R1 

Feature architecture aware phylogenetic profiling indicates a functional diversification of type IVa pili in the nosocomial pathogen *Acinetobacter baumannii*

Dear Dr Ebersberger, 

We are pleased to inform you that your manuscript entitled "Feature architecture aware phylogenetic profiling indicates a functional diversification of type IVa pili in the nosocomial pathogen *Acinetobacter baumannii*" has been formally accepted for publication in PLOS Genetics! Your manuscript is now with our production department and you will be notified of the publication date in due course.

With kind regards,

Jazmin Toth

PLOS Genetics

On behalf of:
